# Identification and Functional Analysis of *APOB* Variants in a Cohort of Hypercholesterolemic Patients

**DOI:** 10.3390/ijms24087635

**Published:** 2023-04-21

**Authors:** Carmen Rodríguez-Jiménez, Gema de la Peña, Javier Sanguino, Sara Poyatos-Peláez, Ana Carazo, Pedro L. Martínez-Hernández, Francisco Arrieta, José M. Mostaza, Diego Gómez-Coronado, Sonia Rodríguez-Nóvoa

**Affiliations:** 1Metabolic Diseases Laboratory, Genetics Department, Hospital Universitario La Paz, Paseo de la Castellana, 261, 28046 Madrid, Spain; crodriguezj@salud.madrid.org (C.R.-J.); jsanguino@salud.madrid.org (J.S.); ana.carazo@salud.madrid.org (A.C.); 2Dyslipidemias of Genetic Origin and Metabolic Diseases Group, Instituto de Investigación Hospital Universitario La Paz (IdiPAZ), Hospital Universitario La Paz, Paseo de la Castellana, 261, 28046 Madrid, Spain; 3Department of Biochemistry-Research, Hospital Universitario Ramón y Cajal, Instituto Ramón y Cajal de Investigación Sanitaria (IRYCIS), Carretera de Colmenar, km 9, 28034 Madrid, Spain; gpenam@salud.madrid.org (G.d.l.P.); sara.poyatos@salud.madrid.org (S.P.-P.); 4Department of Internal Medicine, Hospital Universitario La Paz, Paseo de la Castellana, 261, 28046 Madrid, Spain; pedroluis.martinez@salud.madrid.org; 5Department of Endocrinology and Nutrition, Hospital Universitario Ramón y Cajal, Instituto Ramón y Cajal de Investigación Sanitaria (IRYCIS), Carretera de Colmenar, km 9, 28034 Madrid, Spain; francisco.arrieta@salud.madrid.org; 6Lipid and Vascular Unit, Department of Internal Medicine, Hospital Carlos III-La Paz, Sinesio Delgado, 10, 28029 Madrid, Spain; josemaria.mostaza@salud.madrid.org

**Keywords:** familial hypercholesterolemia, *APOB* variants, low-density lipoprotein, functional assays, co-segregation

## Abstract

Mutations in *APOB* are the second most frequent cause of familial hypercholesterolemia (FH). *APOB* is highly polymorphic, and many variants are benign or of uncertain significance, so functional analysis is necessary to ascertain their pathogenicity. Our aim was to identify and characterize *APOB* variants in patients with hypercholesterolemia. Index patients (*n* = 825) with clinically suspected FH were analyzed using next-generation sequencing. In total, 40% of the patients presented a variant in *LDLR*, *APOB*, *PCSK9* or *LDLRAP1*, with 12% of the variants in *APOB*. These variants showed frequencies in the general population lower than 0.5% and were classified as damaging and/or probably damaging by 3 or more predictors of pathogenicity. The variants *c.10030A>G*;p.(Lys3344Glu) and *c.11401T>A*;p.(Ser3801Thr) were characterized. The p.(Lys3344Glu) variant co-segregated with high low-density lipoprotein (LDL)-cholesterol in 2 families studied. LDL isolated from apoB p.(Lys3344Glu) heterozygous patients showed reduced ability to compete with fluorescently-labelled LDL for cellular binding and uptake compared with control LDL and was markedly deficient in supporting U937 cell proliferation. LDL that was carrying apoB p.(Ser3801Thr) was not defective in competing with control LDL for cellular binding and uptake. We conclude that the apoB p.(Lys3344Glu) variant is defective in the interaction with the LDL receptor and is causative of FH, whereas the apoB p.(Ser3801Thr) variant is benign.

## 1. Introduction

Familial hypercholesterolemia (FH; MIM#143890) is a genetic disorder of lipid metabolism characterized by elevations in low-density lipoprotein (LDL) cholesterol (LDL-c) concentration and a high risk of premature coronary heart disease [1]. The clinical phenotype of FH is most often due to defects in one of 3 genes: *LDLR* (MIM 606945) [1], *APOB* (MIM 107730) [2] or *PCSK9* (MIM 607786) [3]. Among the genetically diagnosed cases, 86–88% are caused by *LDLR* variants, 12% are due to *APOB* variants, and <0.1–2% are produced by *PCSK9* variants [4]. These gene variants can lead to FH with an autosomal dominant pattern of inheritance. The prevalence of the heterozygous form of FH is approximately 1/200–250, and 1/160,000–300,000 in its homozygous form. Subsequent studies have shown that variants of the *LDLRAP1* gene (MIM 605747) result in an autosomal recessive form of FH, which has very low prevalence (1–9/1,000,000) [5,6].

The hypercholesterolemia caused by *APOB* variants is called familial defective apolipoprotein (apo) B-100 (FHCL2, OMIM#144010) [2,7]; these patients have a less severe phenotype than patients with *LDLR* variants [8,9]. The human *APOB* gene is located on the short arm of chromosome 2 (2p24.1), spans 43 kb and includes 29 exons and 28 introns. *APOB* encodes a 4536 amino acid-protein (apoB-100) plus a 27 amino acid signal sequence, expressed mainly in hepatocytes and enterocytes. Exon 26, with 7572 base pairs (bp), encodes more than half of apoB-100 [10]. The *APOB* gene and transcript also encode apoB-48, which coincides with the 48% amino-terminal portion (2152 amino acids) of apoB-100. ApoB-100 mRNA editing involves a specific deamination reaction at nucleotide 6666, in which a cytosine becomes uracil, generating the stop codon UUA and giving rise to the apoB-48 isoform [11,12].

ApoB-100 (~550 kDa), hereinafter apoB, is the major protein component of LDL and the ligand of the LDL receptor (LDLR), with each LDL particle containing a single apoB molecule. The interaction between apoB and LDLR is a key determinant of plasma cholesterol concentration. ApoB has 5 well-defined structural domains, alternating 3 amphipathic alpha-helical domains with 2 amphipathic beta-strand domains, i.e., NH2-βα1-β1-α2-β2-α3-COOH, such that the beta-strands significantly contribute to the high affinity of apoB for lipids [13,14,15]. The LDLR-binding domain is located in the region encompassing residues 3386 and 3396 (3359 and 3369, respectively, not considering the signal peptide), a cluster of basic amino acids within the β2-domain [16]. There are 5 variants in the LDLR-binding domain reported in the Human Gene Mutation Database (HGMD) Professional [4,17,18,19,20], of which only 3 are considered pathogenic, whereas the other 2 are variants of uncertain significance (VUS) [17,18]. However, the first pathogenic *APOB* variant identified and characterized, p.(Arg3527Gln), also known as apoB3527 (formerly apoB3500), is outside this domain. Subsequently, it was shown that Arg3527 interacts with Trp4396 and facilitates the conformation of apoB required for normal binding to the LDLR [21].

The implementation of next generation sequencing (NGS) has expanded gene analysis, reducing time and cost per sample [22]. This innovation has allowed the discovery of a greater number of *APOB* variants, increasing the genetic diagnosis of FH caused by alterations in this gene. However, NGS has also increased the discovery of VUS. There are currently 56 *APOB* variants classified as causing FH by the HGMD Professional. According to the ClinVar database, 58% of reported *APOB* variants associated with FH are currently designated as VUS, whereas only 8% of *LDLR* variants are VUS [23]. To complicate matters further, some missense variants of *APOB* cause hypocholesterolemia or hypobetalipoproteinemia [24]. In contrast, some *APOB* variants have shown incomplete penetrance [4,25,26]. There is, therefore, a need for the functional characterization of variants found in *APOB* to reduce the number of VUS. The aim of this study was to identify *APOB* variants in 825 index patients with clinically suspected FH and perform segregation and functional analyses to discern their pathogenicity. Among the variants found in the canonical genes linked to FH, approximately 12% were located in *APOB*. Functional characterization indicated that the p.(Lys3344Glu) variant is pathogenic, whereas the p.(Ser3801Thr) variant is benign.

## 2. Results

Forty percent of the 825 hypercholesterolemic patients included presented a variant in *LDLR*, *APOB*, *PCSK9,* or *LDLRAP1*. Approximately 12% of these patients showed *APOB* variants with frequencies in the general population lower than 0.5%, classified as damaging and/or probably damaging by 3 or more predictors of pathogenicity (Figure 1). The variant most frequently found was p.(Arg3527Gln), which was present in 6 patients. Two of the *APOB* variants present in patients attended in our centers were selected for functional studies: *c.10030A>G*;p.(Lys3344Glu) and *c.11401T>A*;p.(Ser3801Thr). The variant p.(Ser3801Thr) was found in 4 heterozygous patients and 1 homozygous index patient. This variant is described in some databases as dbSNP and LOVD, and there are conflicting interpretations of its pathogenicity in ClinVar and HGMD Professional 2022.4 [20,27,28]. The variant *c.10030A>G*;p.(Lys3344Glu), which is close to the LDLR-binding domain, was found heterozygous in 2 index cases. This variant was recently reported; however, no co-segregation or functional analyses were performed [20]. The characteristics of the patients and the localization of the variants studied are shown in Table 1.

Moreover, we found 1 patient who had compound heterozygosity for 2 *ABCG8* variants and one patient with double heterozygosity for one *ABCG5* and one *ABCG8* variant. One patient carried a variant in *APOE*, 8 patients in *CYP27A1*, and 2 patients in *STAP1*, whereas none had a *LIPA* variant. Lastly, none of the patients with an *APOB* variant also had a variant in any other FH-related gene.

### 2.1. In Silico Analysis

Both variants, *c.10030A*>*G*;p.(Lys3344Glu) and *c.11401T*>*A*;p.(Ser3801Thr), showed conservation according to Gerp2, and at least 3 of the pathogenicity predictors indicated they were damaging and/or probably damaging (Table 2). According to the American College of Medical Genetics guidelines [29], they were classified as VUS.

### 2.2. Co-Segregation Studies

A co-segregation analysis was possible for the 2 index patients who presented the variant p.(Lys3344Glu): patients 1 and 2. In the case of patient 1, we also studied her son, who had high LDL-c and was a carrier of the variant p.(Lys3344Glu) in heterozygosity, thus showing co-segregation (Figure 2A). Patient 2, besides the p.(Lys3344Glu) variant, carried a second in *APOB*, the *c.8392G>A*;p.(Glu2798Lys) variant. Genetic analysis was possible for 3 other members of the family: 1 brother and 2 nieces of patient 2. These 3 relatives had increased LDL-c and carried the variant p.(Lys3344Glu) in heterozygosity, but lacked the variant p.(Glu2798Lys) (Figure 2B), suggesting that the 2 variants in patient 2 were in trans. Therefore, the results in the 2 families demonstrated co-segregation of hypercholesterolemia with variant p.(Lys3344Glu). Regarding the variant p.(Glu2798Lys), it is new and has not yet been reported in the databases. Given that this variant was absent in the 3 relatives of patient 2, all of who were hypercholesterolemic, co-segregation could not be determined.

No relatives of any index patient carrying the apoB p.(Ser3801Thr) variant were available for the co-segregation analysis of this variant.

### 2.3. Functional Analyses

The conformation of apoB on LDL is critical for LDLR recognition. To assess the binding and uptake of LDL from patients and to preserve the properties of these particles, we chose to analyze their ability to compete with 1,1′-dioctadecyl-3,3,3,3′-tetramethylindocarbocyanineperchlorate (DiI)-labeled LDL, thus avoiding labeling and further manipulation of the patient’s LDL. Whereas the control LDL competed efficiently with DiI-LDL for cellular binding and uptake, LDL from the apoB p.(Lys3344Glu) heterozygous patient 1 was a poor competitor (Figure 3A). To confirm this finding in another carrier of this variant, we analyzed LDL from the son of patient 1 (S1), who was also hypercholesterolemic and heterozygous for apoB p.(Lys3344Glu) (subject II:1 in Figure 2A). These particles, similar to those from his mother, showed a markedly reduced efficiency to compete with DiI-LDL for cellular binding and uptake compared with control LDL (Figure 3B). These results indicate that LDL from the carriers of apoB p.(Lys3344Glu) was defective in the interaction with the LDLR.

In contrast, LDL from patient 3, which bore apoB p.(Ser3801Thr) in homozygosity, competed for uptake as efficiently as control LDL and for binding more efficiently than the latter particles (Figure 3B). These observations suggest that the interaction of such an apoB variant with the LDLR is not defective.

Next, we assessed the interaction of the apoB p.(Lys3344Glu) and p.(Ser3801Thr) variants with macrophages. These cells, while expressing little LDLR, express several scavenger receptors that mediate the uptake of chemically modified LDL, such as oxidized LDL, thereby leading to foam cell formation, a hallmark of atherosclerotic lesions [30]. However, some scavenger receptors, such as CLA-1/SR-BI and CD36, also recognize native LDL [31,32]. The conformational features of apoB also appear to play an important role in this interaction and in the subsequent uptake of LDL by macrophages [33,34]. To ascertain the ability of LDL from patients S1 and 3 to compete with DiI-LDL for macrophage uptake, we used human THP-1 macrophages, which are suitable for assessing non-LDLR-mediated uptake of LDL [35,36,37]. As shown in Figure 4, LDL from both patients competed for macrophage uptake as efficiently as the control LDL did, suggesting that the apoB variants p.(Lys3344Glu) and p.(Ser3801Thr) do not alter the interaction of LDL with macrophage scavenger receptors.

Having observed that LDL that is carrying apoB p.(Lys3344Glu) has a defective uptake through the LDLR, we determined the ability of these particles to provide cholesterol for proliferation of U937 cells, which are known cholesterol auxotrophs, i.e., they depend on an exogenous source of cholesterol to proliferate. To this end, we measured the metabolic incorporation of [methyl-^3^H]-thymidine into cellular DNA in the presence of different LDL. An additional blood sample from patient 3 for this analysis could not be obtained. As shown in Figure 5, LDL from patient S1 was markedly deficient in supporting U937 cell proliferation, and it only approached control values at the highest concentration used. This result is in complete agreement with the inability of LDL from apoB p.(Lys3344Glu) patients to compete for cellular binding and uptake.

## 3. Discussion

*APOB* mutations are a cause of FH; however, the gene is highly polymorphic, with many common and rare variants that do not cause FH. Although more than 1300 unique *APOB* variants have been reported in ClinVar (https://www.ncbi.nlm.nih.gov/clinvar, accessed on 21 February 2023), and 56 have been classified as causing FH in HGMD Professional, there is also a high proportion of VUS [23]. Thus, segregation and functional analyses to classify the *APOB* variants as pathogenic or benign are of utmost importance. In the present study, we performed genetic analyses of 825 index patients with FH and found that approximately 40% carried a variant in one of the main genes causing FH. This result is in agreement with that obtained in cohorts of patients clinically diagnosed with FH worldwide, where a causative variant is found in only 40–50% of cases, though the prevalence of patients with genetically identified FH varies depending on differences in the molecular diagnostic methodologies and clinical criteria applied [4,38,39,40,41]. Approximately 12% of the variants found were in *APOB*, a frequency similar to that previously reported by others [4,42]. Among these variants, we studied the variants p.(Ser3801Thr) and p.(Lys3344Glu).

The variant p.(Ser3801Thr) has been previously reported [20,27,28], and, according to HGMD Professional, it is not a clear cause of FH. We found that this variant displayed a normal interaction with the LDLR in competitive assays. Although further functional and co-segregation studies could not be performed, our results suggest that the variant p.(Ser3801Thr) is benign.

The p.(Lys3344Glu) variant was recently reported [20], but it has been characterized herein for the first time, leading us to classify it as pathogenic and causative of FH. This variant co-segregates with elevated LDL-c, given that all the relatives of index patients 1 and 2 who could be studied had hypercholesterolemia and were p.(Lys3344Glu) carriers. Notably, patient 2 carried a second, novel, apoB variant [p.(Glu2798Lys)]; however, this variant was absent in all her relatives, suggesting that the p.(Lys3344Glu) variant is responsible for the elevated LDL-c in this family.

The p.(Lys3344Glu) variant was defective in the binding and uptake by the LDLR. Interestingly, LDL from the apoB p.(Lys3344Glu) heterozygotes displayed a differential interaction with hepatocytes and macrophages. Although they had deficient binding and uptake by hepatocytes, macrophages took up these particles normally. This outcome suggests that the substitution of Lys with Glu at position 3344 does not alter the interaction of LDL with the scavenger receptors expressed in THP-1 macrophages; however, it specifically affects the recognition by the LDLR. The defective interaction with this receptor is in complete agreement with the inability of LDL that bears apoB p.(Lys3344Glu) to provide cholesterol for cell proliferation.

As mentioned, the apoB p.(Lys3344Glu) variant was only found in heterozygosity. Given that a single apoB molecule is present in each LDL particle, heterozygous patients are expected to have both types of LDL, i.e., LDL carrying mutant and LDL carrying normal apoB. LDL particles with mutant apoB can be expected to have a lower clearance rate than particles with normal apoB, leading to the former particles preferentially accumulating in plasma. This result could explain the negligible proliferation found when LDL from the apoB p.(Lys3344Glu) heterozygous patient was added at low concentrations. The particles carrying normal apoB, which accounted for a lower proportion, could be responsible for the proliferation rate observed at the highest LDL concentration tested.

Position 3344 of apoB is located 42 residues upstream from the LDLR-binding domain (residues 3386 to 3396). The replacement of a basic amino acid with an acidic one (i.e., changing a positive to a negative charge) at this position could secondarily cause a conformational change in the LDLR-binding domain that is incompatible with normal receptor recognition. The variant p.(Ser3346Ile) has been reported 2 amino acids downstream from the 3344 position; however, it has been classified as VUS [18,43]. Many apoB variants causing FH are located outside the LDLR-binding domain [25,28,44,45,46]. The best known are p.(Arg3527Gln) and p.(Arg3527Trp), although the residue replacing this Arg (Arg is also a basic amino acid) does not need to be acidic to disrupt LDLR binding [16,21]. Arg3527 is required to interact with Trp4396, thereby impeding the ability of the C-terminal bow of apoB to interfere with the binding domain [21]. Mutations occurring at 2 other Arg residues in the proximity of Arg3527, p.(Arg3507Trp) and p.(Arg3558Cys), have also been suggested to be important for the correct folding of apoB on LDL [21]. Our results indicate that Lys3344, which is relatively close to the LDLR-binding domain, is crucial to permit the normal interaction of LDL with its receptor, and the apoB p.(Lys3344Glu) variant can therefore be classified as pathogenic.

In summary, NGS enabled us to study all the regions of interest of the FH-causing genes in a large cohort of patients with hypercholesterolemia. We found that 12% of the genetically diagnosed patients carried 1 or 2 *APOB* variants. The p.(Lys3344Glu) variant was identified as causative of hypercholesterolemia, whereas the p.(Ser3801Thr) variant, hitherto classified as VUS, was found to be benign.

## 4. Materials and Methods

### 4.1. Patients

Blood samples from 825 unrelated index patients (20–75-years age range; 372 men and 453 women) with hypercholesterolemia were sent to our reference center. The inclusion criteria for the study were patients clinically classified as having probable or definite FH according to the Dutch Lipid Clinic Network Score. 

### 4.2. Genetic Analysis

The genomic DNA from probands was extracted from EDTA-treated whole blood samples using Chemagen (PerkinElmer, Waltham, MA, USA). DNA was quantified with a NanoDrop ND-1000 Spectrophotometer (ThermoFisher Scientific, Waltham, MA, USA). Genetic analysis was performed by NGS using a customized panel of 435 genes. Library preparation and exome enrichment steps were performed according to the manufacturer’s workflow (Roche NimbleGen, Madison, WI, USA), and sequencing was performed using MiSeq or NextSeq System Sequencing (Illumina Inc., San Diego, CA, USA). A subset of genes was chosen for the analysis: *LDLR*, *APOB*, *PCSK9*, and *LDLRAP1*. In addition, the genes responsible for sitosterolemia, *ABCG5* and *ABCG8*, and genes whose alterations cause FH phenocopies, such as *CYP27A1*, *LIPA*, *APOE,* and *STAP1*, were studied. NGS data were considered suitable after passing the following quality parameters: number of reads more than 30X in 99% of the target bases. Sanger sequencing was used to confirm the presence of the variants found.

### 4.3. Bioinformatic Analysis

Primary analysis was performed using algorithms developed by our Bioinformatics Unit. Briefly, sequences were mapped to the CRCh37/hg19 human reference sequence. The databases used for analysis were HGMD^®^ Professional 2022.4 (https://my.qiagendigitalinsights.com/bbp/view/hgmd/pro/gene.php?gene=APOB, accessed until 31 January 2023), from BIOBASE Corporation; Online Mendelian Inheritance in Man (www.omim.org, accessed until 31 January 2023); and Gene Tests (www.genetests.org, accessed until 31 January 2023). Variant annotation was performed with Ensembl’s Variant Effect Predictor Tool. The analysis software tools used were Trimmomatic v0.36, Bowtie2-align v2.0.6, Samtools v1.3.1, Bedtools v2.26, Picard-tools v1.141, and GenomeAnalysisTK v3.3-0. In silico predictors of pathogenicity used were CADD (Combined Annotation Dependent Depletion), Polyphen (Polymorphism Phenotyping), MutAssesor, Fasthmm, and Vest. Conservation scores were obtained by using Gerp2, PhastCons, and PhyloP in relation to 13 species. The files were uploaded in BAM format for analysis using Alamut Visual V.2.8.0 (Interactive Biosoftware, Rouen, France).

### 4.4. Cell Cultures

HepG2 hepatoma cells (ATCC, HB-8065) were maintained in Dulbecco’s Modified Eagle Medium (DMEM) supplemented with 10% heat-inactivated fetal bovine serum (FBS), 0.1 mM non-essential amino acids, 1 mM sodium pyruvate, 2 mM glutamine, 100 U/mL penicillin, 100 U/mL streptomycin and 10 μg/mL gentamicin at 37 °C in a humidified atmosphere of 5% CO_2_. THP-1 (ATCC, TIB-202) and U937 (ATCC, CRL-1593.2) monocytes were maintained in Roswell Park Memorial Institute (RPMI) 1640 medium containing 10% heat-inactivated FBS and the above-mentioned additives and conditions.

### 4.5. Low-Density Lipoprotein Isolation

Blood was drawn from fasting patients in tubes containing Na_2_-ethylenediaminetetraacetic acid (EDTA) (Vacutainer, Becton Dickinson). Plasma was isolated at 4 °C and treated with 300 kallikrein inhibitor units/mL as a preservative for apoB. LDL (density 1.019–1.063 Kg/L) was isolated by sequential ultracentrifugation and extensively dialyzed against 150 mM NaCl, 0.01% Na_2_-EDTA, pH 7.4, omitting Na_2_-EDTA in the last batch of medium. Protein concentration was determined by a bicinchoninic acid assay (Bio-Rad, Hercules, CA, USA) and cholesterol concentration was measured by an enzymatic method (Centronic GmbH, Wartenberg, Germany). As a control, LDL was always isolated from the same normolipidemic healthy person, who carried no variant in *APOB* or other canonical genes causing FH, and included in each set of functional analyses.

### 4.6. Assays of Competitive Cellular Low-Density Lipoprotein Binding and Uptake

HepG2 cells were plated in 24-well plates (16 × 10^4^ cells per well). After 24 h, cells were pre-treated with 10% lipoprotein-deficient serum (LPDS, density > 1.21 Kg/L) for 24 h to upregulate the LDLR. Subsequently, 20 µg/mL DiI-LDL [32] and increasing concentrations of unlabeled LDL to be tested were added, and the cultures were incubated for 2 h at 4 °C (binding) or 37 °C (uptake). Then, the media were discarded and the cells were washed with phosphate-buffered saline (PBS), trypsinized, resuspended in PBS and analyzed by flow cytometry as described [47].

THP-1 monocytes were differentiated to macrophages with 50 ng/mL phorbol-12-myristate-13-acetate for 48 h in 24-well plates (25 × 10^4^ cells per well). Subsequently, the media were replaced with 10% LPDS, and competitive assays for cellular uptake of LDL and flow cytometry were performed as described for HepG2 cells.

### 4.7. U937 Cell Proliferation Assay

U937 cell proliferation was analyzed by measuring the incorporation of [methyl-^3^H]thymidine into DNA as previously described [48,49]. Briefly, U937 cells were washed in PBS, resuspended in 10% LPDS, and plated in 96-well Multiscreen-HV plates (Millipore, Burlington, MA, USA) at 2 × 10^4^ cells per well. Then, the media were supplemented or not with increasing concentrations of the tested LDL and incubated at 37 °C. At 71 h of incubation, the medium was supplemented with 10 µM 5-fluorodeoxyuridine and 1 h later with 0.5 μCi [methyl-^3^H]thymidine (83.2 Ci/mmol, Perkin-Elmer, Waltham, MA, USA). After a further 18 h incubation, the cells were processed for the measurement of ^3^H.

### 4.8. Statistical Analysis

LDL functionality was analyzed by a 2-way analysis of variance, and post-hoc pairwise multiple comparisons were performed by the Holm-Sidak method.

## Figures and Tables

**Figure 1 ijms-24-07635-f001:**
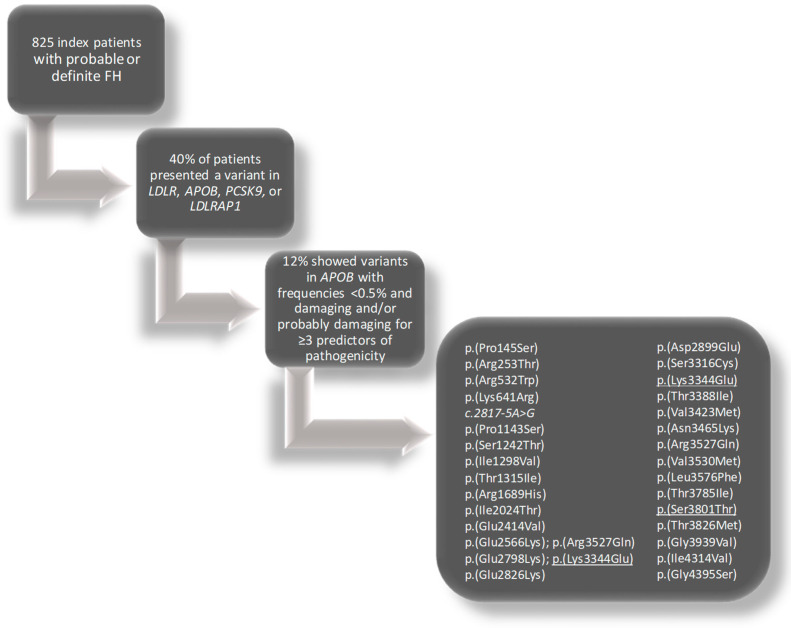
Flow diagram of the study process. The variants characterized are underlined.

**Figure 2 ijms-24-07635-f002:**
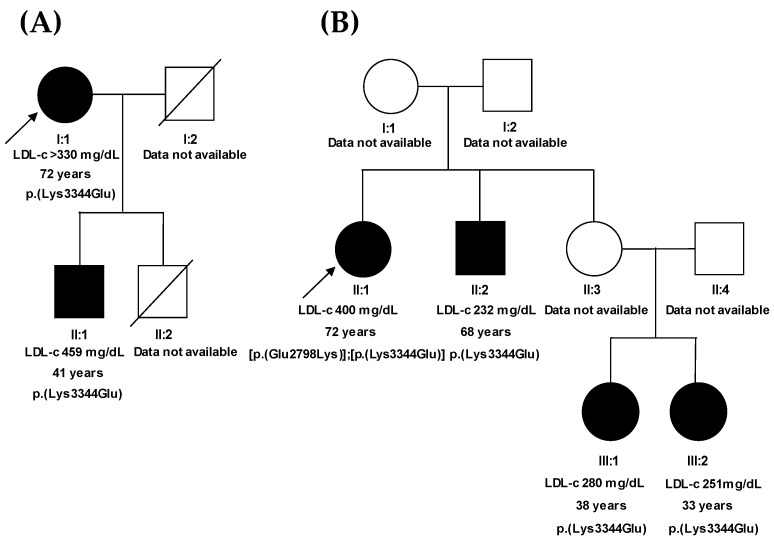
Pedigrees of the families of the 2 index patients carrying the apoB p.(Lys3344Glu) variant. The index cases are indicated with an arrow; circle and square symbols represent women and men, respectively; and shadow-filled symbols indicate the members affected with FH. The first line below the symbols corresponds to the individual identification, the second indicates LDL-c levels before lipid-lowering therapy, the third indicates age, and the fourth indicates the amino acid change in apoB. (**A**) Pedigree of the family of patient 1. (**B**) Pedigree of the family of patient 2.

**Figure 3 ijms-24-07635-f003:**
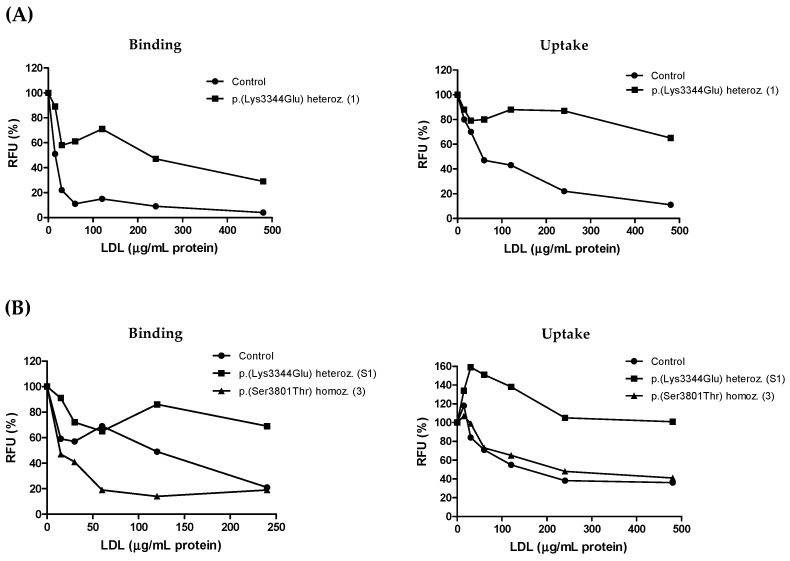
Competitive binding and uptake of LDL in HepG2 cells. (**A**) Ability of LDL from apoB p.(Lys3344Glu) heterozygous patient 1 and the control to compete with DiI-LDL for cellular binding (*p* < 0.001) and uptake (*p* = 0.014). (**B**) Ability of LDL from apoB p.(Lys3344Glu) heterozygous patient S1, apoB p.(Ser3801Thr) homozygous patient 3 and the control to compete with DiI-LDL for cellular binding (patient S1 vs. control, *p* = 0.014; patient 3 vs. control, *p* = 0.023; patient S1 vs. patient 3, *p* < 0.001) and uptake (patient S1 vs. control, *p* < 0.001; patient 3 vs. control, *p* = 0.503; patient S1 vs. patient 3, *p* < 0.001). Results are expressed as percentage of DiI-LDL binding or uptake in the absence of competitor LDL. The identification of each patient is indicated in brackets (see Table 1 and text). RFU, relative fluorescence units.

**Figure 4 ijms-24-07635-f004:**
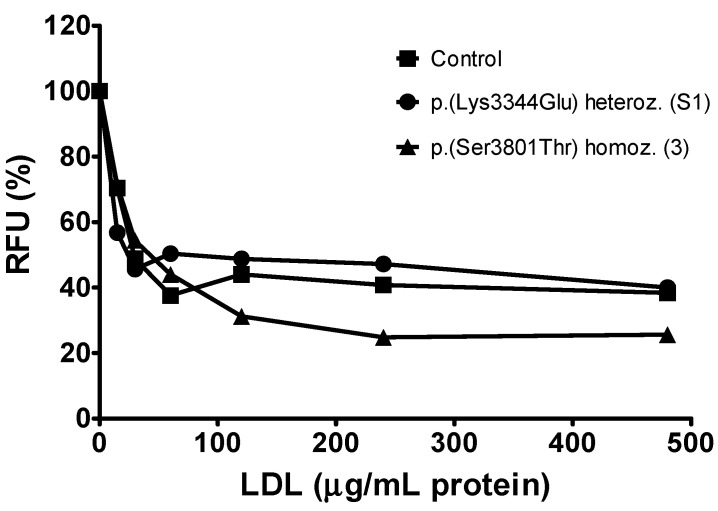
Competitive uptake of LDL by THP-1 macrophages. Ability of LDL from apoB p.(Lys3344Glu) heterozygous patient S1, the apoB p.(Ser3801Thr) homozygous patient 3 and the control to compete with DiI-LDL for cellular uptake (*p* = 0.420). Results are expressed as percentage of DiI-LDL uptake in the absence of competitor LDL. The identification of each patient is indicated in brackets (see Table 1 and text). RFU, relative fluorescence units.

**Figure 5 ijms-24-07635-f005:**
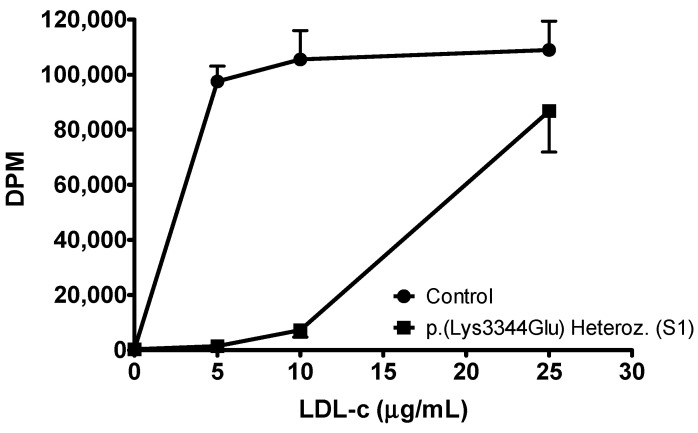
Ability of LDL from apoB p.(Lys3344Glu) heterozygous patient S1 to support U937 cell proliferation. [Methyl-^3^H]thymidine incorporation into DNA was measured in disintegrations per minute (DPM). Data represent the mean ± SEM of four replicates and are representative of two independent experiments with similar results (*p* < 0.001). The identification of the p.(Lys3344Glu) carrier is indicated in brackets (see text).

**Table 1 ijms-24-07635-t001:** Characteristics of the patients and description of the *APOB* variants characterized.

ID	Sex	Age	Origin	Family History		Personal History CVD	Physical Examination		LDL-c (mg/dL)	Exon	Status	cDNA	Protein	Reference
				LDL-c > 210 mg/dL	CVD		Xanthomas	Arcus Cornealis						
1	F	72	Spain	Yes	Yes	No	No	No	≥330	26	Heterozygous	*c.10030A>G*	p.(Lys3344Gln)	[20]
2	F	72	Spain	Yes	No	No	No	No	≥330	26	Heterozygous	*c.10030A>G*	p.(Lys3344Gln)	
3	M	51	Spain	Yes	Yes	No	No	No	220	26	Homozygous	*c.11401T>A*	p.(Ser3801Thr)	[27]
4	F	41	Spain	Unknown	Unknown	Unknown	Unknown	Unknown	155–189	26	Heterozygous	*c.11401T>A*	p.(Ser3801Thr)	
5	F	51	Spain	No	No	No	No	No	190–249	26	Heterozygous	*c.11401T>A*	p.(Ser3801Thr)	
6	M	62	Spain	No	No	No	No	No	250–329	26	Heterozygous	*c.11401T>A*	p.(Ser3801Thr)	

ID: proband identification; CVD: cardiovascular disease; LDL-c: low-density lipoprotein cholesterol.

**Table 2 ijms-24-07635-t002:** In silico analysis of conservation and pathogenicity of the *APOB* variants characterized.

Gene	Variation	Genomic Position	Exon	PhyloP (Vertebrate)	PhastConst (Vertebrate)	Gerp2 Pred	CADD	Sift	Polyphen2	MutAssesor	Fathmm	VEST	Allele Frequency (1000G, GnomAD or ExAC)	ACMG
*APOB NM_000384.2*	*c.10030A>G*;p.(Lys3344Glu)	g.21229710T>C	26	0.964	0.899	Conserved	26.1	Damaging	Damaging	Damaging	Benign	Damaging	Unknown	VUS
*APOB NM_000384.2*	*c.11401T>A*;p.(Ser3801Thr)	g.21228339A>T	26	10.880	0.960	Conserved	22.4	Probably damaging	Damaging	Damaging	Benign	Benign	0.0004	VUS

Predictors of conservation: PhyloP (vertebrate) (score ≤ 2); PhastCons (vertebrate) (range 0–1) and Gerp2 (>2.45 conserved). Predictors of pathogenicity: CADD (s ≥ 14 damaging); Sift (range 0.06–0.23 possibly damaging; Polyphen2 (s ≥ 0.3 damaging); MutAssesor (s < 1.12 benign); Fathmm (range (−1)–(0.08) possibly damaging); and VEST (s ≥ 0.65 damaging). VUS: variant of uncertain significance; ACMG: American College of Medical Genetics.

## Data Availability

The data generated in this study are within the article, available at LOVD (https://www.lovd.nl) or under study.

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
