# Peer review of "Identification and Functional Analysis of APOB Variants in a Cohort of Hypercholesterolemic Patients"

_ijms, 2023, doi:10.3390/ijms24087635_

Round 1
Reviewer 1 Report
The study is interesting in that the authors analyzed VUS in the APOB gene in a large cohort of hypercholesterolemic patients.
1. Have you found rare pathogenic variants in the ABCG5 and ABCG8, and in the genes whose alterations cause FH phenocopies, such as CYP27A1, LIPA, APOE and STEP1, in the cohort of patients with FH?
2. Have any patients, besides the APOB variant, carriers a second variant in the other FH-related genes?
3. Did you use minigene expression analysis to study VUS?
Author Response
Thank you for your comments.
- Have you found rare pathogenic variants in the ABCG5and ABCG8, and in the genes whose alterations cause FH phenocopies, such as CYP27A1, LIPA, APOE and STEP1, in the cohort of patients with FH?
Response: We found some variants in most of the genes you mention. This has been included in an additional paragraph in the first part of Results (second paragraph, page 5), as follows: “Moreover, we found one patient who was compound heterozygote for two ABCG8 variants and one double heterozygote for one ABCG5 and one ABCG8 variant. One patient carried a variant in APOE, eight patients in CYP27A1 and two patients in STAP1, while none had a LIPA variant”.
- Have any patients, besides the APOBvariant, carriers a second variant in the other FH-related genes?
Response: None of the patients with an APOB variant carried also a variant in any other FH-related gene, as has been stated in the last sentence of the new paragraph added to the first part of Results (second paragraph, page 5), as follows: “Finally, none of the patients with an APOB variant also had a variant in any other FH-related gene”.
- Did you use minigene expression analysis to study VUS?
Response: We have not used minigenes to study VUS. The APOB variants functionally studied in this paper mapped in exon regions and were missense variants that were analyzed by isolating LDL. The use of minigenes seems to be developed for the validation of stop or frameshift variants in APOB that produce alterations in splicing, generating different transcripts.
Reviewer 2 Report
The authors identified and performed the functional analysis of two ApoB mutants from FH patients with in vitro experiments. The cosegregation analysis showed that hyperlipidemia is correlated with p. (Lys3344Glu). LDL particles from patients carrying Lys3344Glu showed decreased binding capacity to HepG2 cells, leading to lower cell uptake. In comparison, the authors argued that the mutant of Ser3801Thr did not show decreased ability for cell binding and uptake. The results are interesting; however, several concerns need to be resolved before publication.
1. The information for cosegregation analysis in Fig 2 is not completed, which needs to be further discussed in the Discussion.
2. Have the authors performed the cosegregation analysis with Ser3801Thr?
3. Figure 3: Why the patterns for controls of uptake in Fig 3a and Fig 3b are so different? What is the difference between the sample of p.(Lys3344Glu) heteroz (1) in Fig3a and the sample of p(Lys3344Glu) heteroz (S1) in Fig3b? Why there are differences in binding and uptake for those two samples? In the text (lines 161-162), the author stated that “LDL from patient 3, which bears apoB p.(Ser3801Thr) in homozygosity, competed for 161 binding and uptake as efficiently as control LDL (Figure 3b)”. However, Fig 3a showed that the uptake was similar to controls, but the binding is obviously different. Information on statistical analysis should be provided in the figure legend.
4. Figure 5: The results need to be described in more detail; a brief introduction of methods might be needed for this figure. For example, why the DPM represents the U937 cell proliferation in Fig 5? Why did you choose U937 cells for proliferation? Statistical analysis information is required in the figure legend.
Author Response
Thank you for your comments and suggestions.
- The information for cosegregation analysis in Fig 2 is not completed, which needs to be further discussed in the Discussion.
Response: The results of the cosegregation analysis (of the p.(Lys3344Glu) variant) have been further discussed in the third paragraph of Discussion. For this, the beginning of the fourth paragraph has also been modified.
- Have the authors performed the cosegregation analysis with Ser3801Thr?
Response: Unfortunately, it was not possible to perform the cosegregation analysis with the Ser3801Thr variant. For reasons beyond our control, it was not possible to study any relative of the index patients carrying this variant. In the 2nd paragraph of Discussion, it was already stated that “… cosegregation studies could not be performed” for this variant, but to make this clearer, a new sentence has now been added in Results (Section 2.2. Cosegregation studies, last sentence) as follows: “No relatives of any index patient carrying the apoB p.(Ser3801Thr) variant were available to perform the cosegregation analysis of this variant”.
- Figure 3: Why the patterns for controls of uptake in Fig 3a and Fig 3b are so different? What is the difference between the sample of p.(Lys3344Glu) heteroz (1) in Fig3a and the sample of p(Lys3344Glu) heteroz (S1) in Fig3b? Why there are differences in binding and uptake for those two samples? In the text (lines 161-162), the author stated that “LDL from patient 3, which bears apoB p.(Ser3801Thr) in homozygosity, competed for 161 binding and uptake as efficiently as control LDL (Figure 3b)”. However, Fig 3a showed that the uptake was similar to controls, but the binding is obviously different. Information on statistical analysis should be provided in the figure legend.
Response: The reason for the difference between the patterns of the control is unknown. It should be taken into account that the two studies, a and b, presented in Fig. 3 were performed at different times, using different blood samples to isolate fresh LDL from the control individual. Obviously, the corresponding cellular assays were also performed independently. These facts may introduce some variability in the values of the data obtained, but, even then, the clear difference in the biological properties between control LDL and LDL carrying apoB p.(Lys3344Glu) is evident in both assays.
As commented in the text, patient S1 is the son of index patient 1, and both are heterozygous for apoB p.(Lys3344Glu). Patient S1 was studied to confirm the defective interaction of LDL carrying this apoB variant, as previously found in his mother. Again, LDL from mother and son were isolated and assayed at different times, and it is difficult to compare the results directly (let alone that LDL particles are considerably complex and heterogeneous in composition and size), but, importantly, these results were consistent with the defective cellular binding and uptake of LDL bearing apoB p.(Lys3344Glu) as compared to control LDL.
In Fig. 3b, the competitive curves for binding of LDL carrying apoB p.(Ser3801Thr) and control LDL are different (P = 0.023), with the former particles competing more efficiently than control LDL. However, the competitive curves for uptake were similar. These facts, which also support that the patient’s LDL is not functionally defective, has been clarified more precisely by amending the second paragraph of section 2.3. Functional analyses as follows: “On the other hand, LDL from patient 3, which bears apoB p.(Ser3801Thr) in homozygosity, competed for uptake as efficiently as control LDL, whereas it competed for binding more efficiently than the latter particles (Figure 3b). These observations suggest that the interaction of such an apoB variant with the LDLR is not defective”.
Information on statistical analysis is now provided in the figure legend. The section 4.8. Statistical analysis has been added to Methods. Moreover, in the legend to the figure it is specified that the numbers in the brackets indicate the identification of the patient studied.
- Figure 5: The results need to be described in more detail; a brief introduction of methods might be needed for this figure. For example, why the DPM represents the U937 cell proliferation in Fig 5? Why did you choose U937 cells for proliferation? Statistical analysis information is required in the figure legend.
Response: Metabolic incorporation of [methyl-3H]thymidine into cellular DNA is a widely used and sensitive method to monitor DNA synthesis and cell proliferation. U937 cells are cholesterol auxotrophs, i.e. they depend on exogenous cholesterol provision to proliferate. The ability of LDL as an exogenous source of cholesterol for cell proliferation is often used as a test to assess the functionality of these particles. Therefore, we measured the metabolic incorporation of [methyl- 3H]thymidine into cellular DNA in the presence of tested LDL. To better describe this approach, we added the following sentences to the last paragraph of 2.3. Functional analyses: “…, i.e. they depend on an exogenous source of cholesterol to proliferate. For this, we measured the metabolic incorporation of [methyl-3H]-thymidine into cellular DNA in the presence of different LDL”. Moreover, the following sentence was added to the legend of Figure 5: “[Methyl-3H]thymidine incorporation into DNA was measured in DPM”.
Information on statistical analysis is now provided in the figure legend. The section 4.8. Statistical analysis has been added to Methods.